# The Production of Three-Dimensional Metal Objects Using Oscillatory-Strain-Assisted Fine Wire Shaping and Joining

**DOI:** 10.3390/ma17102188

**Published:** 2024-05-07

**Authors:** Anagh Deshpande, Keng Hsu

**Affiliations:** 1Reverb Industrial, San Leandro, CA 94577, USA; anagh.deshpande@gmail.com; 2School of Manufacturing Systems and Networks, Arizona State University, Mesa, AZ 85212, USA

**Keywords:** metal 3d printing, metal additive manufacturing, aluminum printing

## Abstract

Material shaping and joining are the two fundamental processes that lie at the core of many forms of metal manufacturing techniques, including additive manufacturing. Current metal additive manufacturing processes such as laser/e-beam powder bed fusion and Directed Energy Deposition predominantly use heat and subsequent melt–fusion and solidification to achieve shaping and joining. The energy efficiency of these processes is severely limited due to energy conversion losses before energy is delivered at the point of melt–fusion for shaping and joining, and due to losses through heat transfer to the surrounding environment. This manuscript demonstrates that by using the physical phenomenon of lowered yield stress of metals and enhanced diffusion in the presence of low amplitude high frequency oscillatory strain, metal shaping and joining can be performed in an energy-efficient way. The two performed simultaneously enable a metal additive manufacturing process, namely Resonance-Assisted Deposition (RAD), that has several unique capabilities, like the ability to print net-shape components from hard-to-weld alloys like Al6061 and the ability to print components with a very high aspect ratio. In this study, we show this process’s capabilities by printing solid components using aluminum-based metal alloys.

## 1. Introduction

Material shaping and joining are the two fundamental processes that lie at the core of many forms of metal manufacturing operations: formative processes like forging, additive processes like laser or electron-beam powder bed fusion, and welding operations like Metal Inert Gas (MIG) welding. Fundamentally, metal shaping, or permanent/plastic deformation, begins when crystalline defects like dislocations in metals start to move along their slip planes. Joining, however, involves the exchange of mass across boundaries or interfaces between adjacent domains of metal. Mass transfer processes such as diffusion or advective mixing drive the fusing of interfaces in joining. Shaping and joining both require energy input to overcome the initial energy barrier and to sustain the continuation of these processes. In addition to the energy input used to drive these processes, additional energy can be used to assist these processes to allow for larger amounts of deformation, or to accelerate the diffusion behavior. Several forms of energy can be used to assist metal shaping and joining. In the case of metal shaping, many operations utilize heat energy to reduce hardening and to allow the plastic deformation, or shaping, of metals at flow stresses lower than the static yield strength of the material. Similarly, in joining, many processes use heat energy to create a molten pool of metals that is then joined as a result of the mass transfer happening in the melt pool. These fundamental processes are performed independently or sequentially to fabricate components and assemblies in manufacturing processes [1].

Additive manufacturing processes are no exception. In additive manufacturing, shaping and joining are performed either simultaneously or sequentially to produce components. In the currently available metal additive manufacturing processes, this can be accomplished in a few ways. First, the process can use a high-power “point” heat source such as a focused laser or an electron beam following a prescribed path to create a melt pool from metal powder or wire feedstock, which fuses and subsequently solidifies to form solid objects line by line and layer by layer. Examples technologies that use this process include the Selective Laser Melting (or laser powder bed fusion) and Directed Energy Deposition techniques [2]. Another method involves depositing metal particles with the help of a polymer binder, which is then followed by steps to burn off the binder and sinter the metal particles together in the follow-up post processes. Examples of this type of process include the Binder Jetting additive manufacturing, Bound Metal Extrusion, and Metal Nano-Particle Jetting additive manufacturing technologies [3,4]. Yet another possible approach taken to perform metal additive manufacturing involves the use of localized frictional heat fusion and mechanical strain to shape and join feedstock material into solids. Examples of this include the Ultrasonic additive manufacturing and Friction Stir additive manufacturing technologies [5,6]. While these processes have proven capabilities and applications, fundamentally, the energy efficiency of these processes is severely limited because they rely on several energy conversion processes between energy input to material shaping and joining. Laser powder bed fusion, for example, uses solid-state fiber lasers that operate at a low energy efficiency. The laser energy incident into the material then goes through another absorption loss due to metal absorptivity issues in both solid and liquid states. At this point, the absorbed energy further experiences losses due to heat transfer from the laser incident zone into its surroundings before it serves the purpose of heating up the metal feedstock and forming the melt pool to accomplish shaping and joining [7]. Additionally, the inefficiencies in the conversion process from electrical energy to optical energy cause parasitic heating, which, in turn, requires yet additional energy to dissipate in order to mitigate lasing issues. As a result, a laser powder bed fusion AM machine typically consumes 10–20 kW of power during operation, which translates to 300–500 MJ/Kg (or 1 MJ/cm^3^) of energy consumption for processing metals like aluminum alloys at the machine level [7,8]. These energy inefficiency issues are not any better in other thermal or melt–fusion-based technologies. 

In contrast, when one considers the findings reported by Langenecker [9], the use of high-frequency oscillatory strain to directly couple mechanical energy into lattice structures and the defects within crystalline metals to assist deformation represents a significantly more energetically efficient approach to metal shaping and joining. These findings clearly indicate that using small-amplitude high-frequency oscillatory strain imposed in metals to assist in shaping results in a significant drop in the metal’s apparent yield and flow stress. Energetically speaking, using small-amplitude (less than a few tens of micrometers) repeated strains (tens of kHz) is roughly 30 times more efficient than using heat to cause the same amount of yield or flow stress reduction in metal shaping, under theoretical conditions with no heat lost or heat transfer from a 1 cm^3^ block of aluminum [9].In reality, the energy efficiency benefits are likely more than a factor of 30 because of the heat transfer and therefore losses during heating.

The solid-state joining of metal surfaces fundamentally requires the physical advective mixing or atomic diffusion of metals across interfaces. Inherently, diffusion rates are higher at metal surfaces or grain boundaries due to the increased amount of surface diffusion. The conventional approach of using elevated temperatures to accelerate solid-state diffusion is an intuitive way to increase diffusion rates since the diffusivity of self-diffusion of metals increases with temperature. As compared with how diffusion is enhanced by the presence of crystalline defects such as dislocation pipes, however, mass transport in high crystalline defect regions is more effective [10]. As demonstrated in Legro’s work, where the bulk diffusion of aluminum in aluminum is compared with aluminum diffusion in dislocation pipes, increasing the temperature of aluminum from 573 K to 673 K allows for a roughly 100-fold increase in diffusivity to be observed. In contrast, at 573 K, the diffusivity of aluminum along a lattice line defect like dislocation is nearly 10,000 times higher than that observed in bulk diffusion [10]. Conceptually, this means that if one could induce large amounts of lattice defects in the vicinity of metal–metal interfaces with pristine contact, along with a defect concentration gradient across the interface, enhanced diffusion across said interface, and therefore joining, would be expected. Taken together, the enhancement in the ability to shape metal using energetically more efficient oscillatory vibrations and the increased solid-state mass-transfer-based joining can form the basis of a method to simultaneously shape and join small units of metal material without melting or ever heating it. As this method is implemented in a way to gain control of the unit metal shaping and joining process both in space and time, an entirely new 3D-printing approach can emerge.

In this manuscript, a novel method, namely the Resonance-Assisted Deposition (RAD) method, for the 3D printing of metals by simultaneously shaping and joining a fine wire continuously into metal voxels in a solid state is demonstrated. It is shown that this method is significantly more energy-efficient than using heating, melting, and solidification. This method has the ability to print net-shape components from hard-to-weld alloys like Al6061 and the ability to print components with a high aspect ratio. Because of these specific advantages, this novel method has the potential to become the method of choice in applications where existing processes for metal 3D printing are lacking.

## 2. Methodology

### 2.1. Resonance-Assisted Deposition Printing

To implement the RAD (Resonance-Assisted Deposition) method into a system to produce a 3D metal object, a 3-axis motion system was used. A schematic of the experimental setup is shown in Figure 1a. The process begins with slicing of a CAD model into layers and then subsequent generation of toolpaths to fill the layers. The toolpaths are then sent to the motion system for execution. The build plate is allowed to move in the X-Y plane and the print-head is allowed to move in the Z-direction. Solid wire feedstock is made to pass through a hollow cylindrical shear-strain transfer tool that has a hole of 0.5 mm diameter. The tool is attached to a transducer driven by a piezo crystal which vibrates at a frequency of 40 kHz. In our RAD process for metal 3D printing, each compression cycle is the basic step that produces a voxel. When arranged in a desired geometry, these voxels form a net-shape 3D component. In a compression cycle, the feedstock is brought to a desired location. The downward motion of the print head transfers the oscillatory shear strain through the shear-strain transfer tool to the filament. As described in the previous section, the transfer of this oscillatory shear strain to the filament lowers the yield stress of the filament and allows for the shaping of the voxel being deposited. The shape of the voxel then changes from a circular cross section to a rectangular (slab-like) cross section, as shown in Figure 1b. At the same time, the oscillatory shear strain also allows for the joining of the voxel to the lower layer and the adjoining material because of the enhanced diffusion across the material interfaces. Once the voxel has been shaped and deposited, the shear-strain transfer tool then lifts up and moves along the X–Y plane by a certain “step size”, and the compression cycle repeats to deposit another voxel. Figure 1c shows the schematic of the tool motion during one compression cycle. Several such deposited voxels form a layer of deposited material. These layers when stacked on top of each other form a net-shape 3D component, as shown in Figure 2. Movie 1 (Appendix A) shows a real-time video clip of the process.

All articles displayed in Figure 2, Figure 3, Figure 4, Figure 5 and Figure 6 were printed with annealed Al6061-O feedstock wire 0.350 mm in diameter supplied by the California Fine Wire Company (Santa Barbara, CA, USA). The feedstock wire had a tensile strength of 140 MPa and an elongation of 17.1%. The tool pathing strategy used for all of the prints is shown in Figure 5a. The distance by which the capillary tool moves to deposit the next voxel (called the step size) was 1 mm. Experiments were conducted with an inter-track distance of 0.6 mm and 0.7 mm to achieve space filling and eliminate inter-track voids. Table 1 summarizes the process parameters used.

### 2.2. Surface Roughness, Voids, and Defect Analysis

To analyze the effect of process parameters on the voids and defects in printed components, a cross section of a printed component was cut, and standard metallographic polishing was performed on the cross section. Post metallographic polishing, optical images of the polished surface were captured to reveal voids and other defects, as shown in Figure 5. Micro-CT analysis was performed to analyze the spatial location, distribution, and overall dimensions of defects in the printed components. For this purpose, Thermo Scientific HeliScan Mk2 was used and the scans were performed at a resolution of 6.95 μm. 

To analyze the average surface roughness of the printed components, a Mitutoyo Surftest portable surface roughness tester was used. The surface roughness was measured on surfaces parallel to the X–Y and X–Z (or Y–Z) planes.

### 2.3. Tensile Testing

Tensile testing was performed on test coupons printed along the print direction and orthogonal to the print direction in an Instron 120 Tensile testing machine at a constant speed of 150 mm/min. The dimensions of the printed coupons are given in Appendix A of the Appendix A. Engineering stress and strain values were recorded and plotted.

### 2.4. Energy Consumption Measurement

The oscillatory strain generator, motion, and control system (Figure 1) used to demonstrate metal 3D printing using RAD was outfitted with a power meter (KW47-US), which measures the power draw of the entire system during printing. The power consumption of the entire system was monitored and logged for comparison with other metal additive manufacturing processes.

### 2.5. TEM Analysis

To analyze the diffusion across the interfaces during voxel deposition, pure Al (99.99%) voxel was deposited on pure Nickel to eliminate any effect of alloying elements in the analysis. Two different metals were used for these experiments so that the Energy-Dispersive Spectroscopy (EDS) technique could be used to quantitatively measure the diffusion across the interface. Focused Ion Beam was used to prepare an electron-transparent lamella of the interface, which was analyzed using a Tecnai G2 F20 200 kV TEM equipped with an EDAX EDS detector. High-resolution TEM images and EDS line scans were captured. For EDS, an accelerating voltage of 200 kV, a spot size of 2, and a step size of 5 nm were used. For TEM imaging, an accelerating voltage of 200 kV and a spot size of 1 were used.

## 3. Results and Discussion

### 3.1. Shaping and Joining of Volumetric Units (Voxels) to Form a 3D Object

During the voxel deposition process described in Section 2.1 (the Resonance-Assisted Deposition Printing section), the imposed oscillatory shear strain has two effects on the voxel material being deposited that assist its shaping and joining. The first is that it lowers the yield stress of the material so that the voxel can be shaped at stresses much lower than the yield stress of the material. This effect has been extensively modeled and studied [11,12,13,14]. The oscillatory shear-strain energy enables the faster annihilation of dislocations, resulting in reduced hardening and the lowering of flow stress required to plastically deform the metal [14]. The oscillatory shear strain imposed on the voxel also causes friction at the feedstock–substrate interface (or the feedstock–lower-layer interface), but only during the initial few milliseconds. Because the voxel initially has a circular cross section, the area of contact between the voxel and the surface of the stiff capillary tool applying the oscillatory strain and compressive force is small, allowing for relative movement between the interfaces, including the voxel–material interface. As the voxel is compressed and shaped, the area of contact increases continuously until a point is reached when the relative movement between the surfaces stops, and the voxel experiences purely oscillatory shear strain. It is at this point that frictional heat on the interfaces ceases to be generated. The frictional heat that was generated in the initial few milliseconds is not sufficient to cause a significant increase in the temperature of the voxel, as shown in our previous published work (Appendix A, Appendix A) [15]. However, the combination of this friction and the increase in surface area because of the shape change allows the breaking and redistribution of native oxides on the surface of the wire feedstock, bringing about nascent metal-to-metal contact at the interface.

Simultaneously, as the voxel is being compressed and shaped, the second crystalline defect and oscillatory-shear-strain-based phenomenon promotes atomic diffusion across the interface, which allows the joining of the voxel to other neighboring ones. This athermal diffusion enhancement phenomenon has also been employed in the wire bonding process for applications in semiconductor device packaging [16]. It is hypothesized that the chemical potential that is required for atomic diffusion is raised across voxel–voxel boundaries because of the crystalline defect density gradient across the same interfaces, as illustrated by Legro et al. [10]. This gradient in RAD is a result of the high-defect-density region formed during the deposition process. The interface of voxels deposited using RAD was analyzed by conducting experiments in which pure aluminum voxels were deposited on a nickel substrate and the interface was studied using TEM. This quantitative analysis of the Al-Ni system will not directly apply to the Al-Al system typically seen in RAD because the diffusivity of Al in Ni is different from the diffusivity of Al in Al. The main objective of our Al-Ni investigation was to provide evidence of enhanced diffusion across the interface during the deposition process in RAD. To conduct this investigation, it was essential to use some material other than Al as a substrate so that the concentration of the two metals across the interface could be measured using EDS. 

The high-defect-density region that is hypothesized to increase the chemical potential is evident in Figure 3a, which shows a TEM bright-field image of the Al-Ni interface. Native oxides that break due to the initial friction get trapped at the interface during the deposition process and can also be seen in Figure 3a,c. The high-resolution TEM image in Figure 3b reveals that the defects were actually stacking faults. To quantify this enhanced diffusion phenomenon during RAD printing and to compare it with heat-assisted diffusion, EDS line scans were taken at the Al-Ni interface. The results and the associated calculations are given in the Appendix A section, which provides further analyses of oscillatory-strain-assisted enhanced diffusion. The width of the diffusion zone was found to range between ~80 nm and ~140 nm. If this diffusion were to be attributed purely to a heat-assisted enhancement in diffusion, we found that the effective constant temperature that would be required at the interface ranged from ~320 °C to 420 °C (see Appendix A). In our previously published work [14] regarding a precursor method to this RAD technique that uses the same deposition method, thermocouple and high-speed infrared camera measurements (Appendix A, Appendix A) showed that the temperature rise actually measured during the deposition process was in the order of 5–10 °C. This indicates that the temperature rise cannot account for the observed diffusion and that oscillatory shear strain does assist in promoting atomic diffusion across the voxel interfaces to join each voxel to the neighboring ones.

### 3.2. Surface Morphology

As seen in Figure 4b,c, the top surfaces of the printed metal parts (surfaces orthogonal to the print direction) have different surface morphologies as compared with the side surfaces parallel to the build direction. The top surface morphology is dictated by the tooling surface in contact with the metal voxel during each compression cycle, as well as the material flow and track-to-track overlaps in the tool path. The bottom surface morphology of the build is determined by the joint strength between the first layer of the printed part and the build plate, and how it is separated from the build plate. The side surfaces have a characteristic “scalloping” effect, similar to what is typically seen in the Fused Filament Fabrication polymer 3D-printing method, as a large amount of material strain is experienced by the feedstock material during the printing process. In the RAD process, the ratio of voxel height (also the layer height) to feedstock diameter is around 0.4. While the exact ratio is dictated by the amount of compression required in each compression cycle to establish full space filling between adjacent tracks and the proper joining of interfaces, this ratio also determines the extent of the scallops on the side surfaces. The larger the layer height, the higher the waviness on the sidewalls. Within the feasible parameter range, the resulting roughness values of the top surfaces were typically around 10–20 μm Ra, while the side surfaces were in the range of 15–25 μm Ra.

### 3.3. Energy Density Input Calculations

The energy input of the RAD technique reported here is mechanical strain energy in nature. The total energy input into the material during voxel compression shaping and joining is equal to the sum of the shear-strain energy in each high-frequency shear-strain cycle and that of the semi-static compression in the orthogonal direction (Z-direction). This can be expressed as
$$W_{total}=F_{osci}\delta_{osci}+F_{static}\delta_{static}$$
where $F_{osci}$ is the force required to shear the voxel in the direction of the oscillatory strain;
$\delta_{osci}$ is the oscillatory shear displacement;$F_{static}$ is the force required to compress the voxel in the Z-direction;$\delta_{static}$ is the compression of the voxel to its layer height.

Substituting $\delta_{osci}\;\mathrm{and}\;\;\delta_{static}\;$with the actual process parameters of the tool vibration and static compression displacements, respectively, while replacing the forces experienced by the material in each vibration cycle, $F_{osci}$, with the product of the modulus and shear displacement, yields
$$W_{total}=GA\frac{{\left(4aft\right)}^{2}}{h}+F_{static}\left(d-h\right)$$
where *G* is the bulk modulus of the wire feedstock material, *A* is the voxel X–Y plane area of the voxel being deposited, *a* is the vibration amplitude in each oscillating shear-strain cycle, *f* is the vibration shear-strain frequency, *t* is the voxel compression time, *d* is the wire feedstock diameter, and *h* is the height of the voxel (or the layer height). For the aluminum 6061-O material described in this work, the process parameters relevant to this energy density calculation are listed in Table 2.

Based on these input parameters, the total energy density input used in shaping and joining a voxel of material is estimated to be 3.79 × 10^−4^ J/mm^3^ for a vibration frequency of 40 kHz. Note that in these calculations, the voxel (layer) height is assumed to be the final height after the voxel is completed for estimation of the upper-bound value. The same assumption is also used for A, the voxel plane area, where the final area of the voxel is used. 

In the context of the energy consumption and efficiency of technological processes, however, power consumptions at the machine system level would provide a better measure of the overall energy efficiency and consumptions of this RAD technique as a 3D-printing technological tool for metal manufacturing. To this end, the entire system (Figure 1) used to demonstrate metal 3D printing using RAD was outfitted with a power meter that measured the power draw of the entire system during printing. At the machine system level, the system consumed on average 100–300 Watts of electrical power during printing, depending on whether build plate heating was used to assist in the in-process annealing of the printed material. In comparison with typical LPBF or DED metal AM systems, a system that utilizes the RAD technique for metal 3D printing requires less power by roughly a factor of 10–100 [7,17].

### 3.4. Mechanical Properties

In each compression cycle, the feedstock is compressed in the build direction to the voxel height as it expands laterally to form the voxel width. If this width is more than the spacing between the parallel tracks (called the inter-track distance), complete space filling of the cross-section is achieved. For an inter-track distance of 0.7 mm, the lateral flow of material was insufficient to completely fill the inter-track space, resulting in voids. The resulting voids are shown in Figure 5b,e. Shown in Figure 5c is an example of cross-sectional space filling with proper overlapping, which resulted from an inter-track distance of 0.6 mm. Under these conditions, we observe that the lateral flow of material enables the spaces between the track of metal and its neighboring tracks to be completely filled, and the overall density of the build part reaches 99.95% or greater of that of the feedstock material (micro-CT results in Figure 5e). The remaining defects seen in Figure 5e were predominantly found between the outer wall and infill, as shown in Figure 5a. In the cross sections shown in Figure 5b,c, darker boundaries clearly delineate the inter-layer interface. This is because of the native oxides discussed earlier (Figure 3a) that became trapped at the inter-layer interface, preventing nascent metal contact.

The mechanical properties of the parts built using this RAD technique reflect very low anisotropy in ultimate tensile strength but a higher anisotropy in elongation (Figure 6a). The ultimate tensile strength of the as-printed samples was found to be close to 75% of that of the raw Al-6061 O-annealed feedstock. The lower elongation can be attributed to the trapped native oxides discussed in the previous section, which result in easier crack propagation in the regions of the interface where the oxides have been trapped. In the fracture surfaces shown in Figure 6, one can clearly observe the extensive plastic deformation in the regions between the layer-to-layer interfaces, while the interface regions exhibit significantly lower ductility. On the other hand, the overall fracture behavior of the built material exhibited less ductility when a tensile load was applied along the build direction, exhibiting a brittle fracture, as is evident in the fracture surface shown in Figure 6f, which shows no evidence of plastic deformation but shows just a clear separation at the inter-layer interface. 

## 4. Conclusions

This work demonstrates the Resonance-Assisted Deposition technique, wherein high-frequency small-amplitude oscillatory strains are used to shape and join segments of a solid metal wire into tracks and layers of enclosed areas of metal material. Combined with the ability to control the timing and positioning in space of such shaping and joining, RAD is shown to be capable of producing three-dimensional metal objects without heating or melting the metal feedstock. As compared with conventional melt–fusion-based techniques for metal 3D printing, such as the laser powder bed fusion (LPBF) technique, the energy efficiency of our RAD technique at the voxel level is estimated to be orders of magnitude higher (100 J/mm^3^ for LPBF, less than 0.1 mJ/mm^3^ for RAD). As-built 6061 aluminum-O objects produced using this RAD technique are shown to have a 99.95% as-built specific density. The strength of these as-built parts was within 75% of that of the feedstock material, with the elongation being lower for parts when tensile load was applied in the build direction. With its high material- and system-level energy efficiency, and near-net-shape capability, this Resonance-Assisted Deposition technique has the potential to bring about point-of-need metal additive manufacturing in resource limited or extreme environments, such as in space, at sea, or during transportation.

## Figures and Tables

**Figure 1 materials-17-02188-f001:**
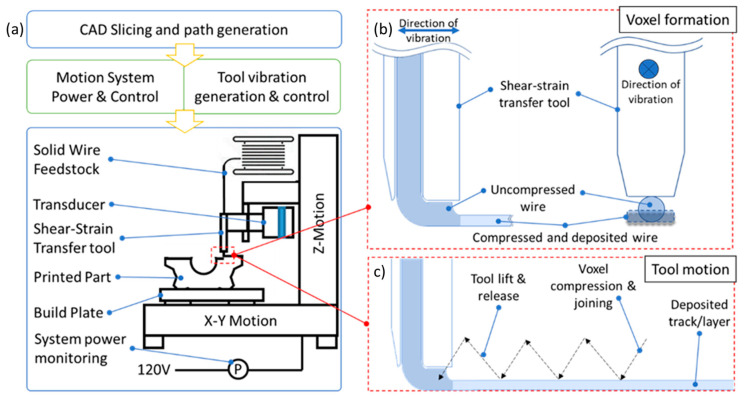
In this figure, (**a**) shows a schematic of the experimental setup for Resonance-Assisted Deposition (RAD) 3D printing, (**b**) shows a representation of the voxel formation process, and (**c**) shows the tool motion during the RAD process.

**Figure 2 materials-17-02188-f002:**
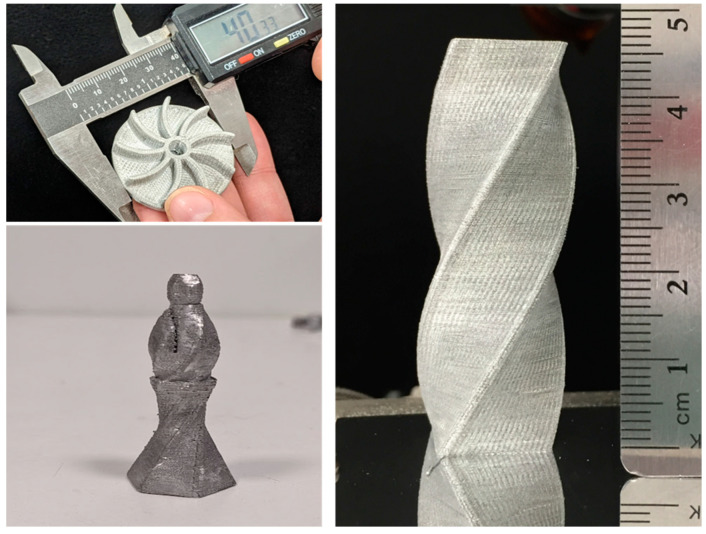
Aluminum 6061 alloy objects fabricated using the RAD process.

**Figure 3 materials-17-02188-f003:**
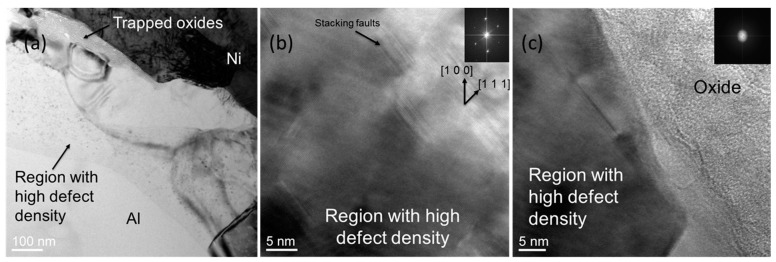
In this figure, (**a**) shows a bright field TEM image of the interface of pure Al and pure Ni joined using the RAD technique. As mentioned in the manuscript, the initial oscillatory strain breaks the oxide layer from Al. In some cases, the oxide becomes trapped in the inter-layer interface, as shown in the image. A region near the interface with a high density of crystalline defects in Al is clearly seen in this image. High-resolution TEM images of this region with a high defect density are shown in (**b**,**c**). The crystalline defects (stacking faults) are apparent in (**b**). The trapped oxide, which is amorphous in nature, as evident by the diffraction pattern, is shown in (**c**).

**Figure 4 materials-17-02188-f004:**
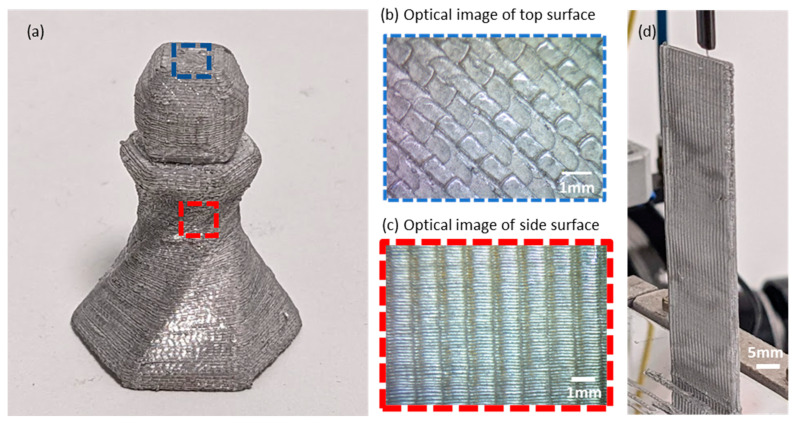
(**a**) An Al6061 article printed with the RAD process; (**b**,**c**) optical images of surfaces parallel to the XY plane and the XZ plane, respectively, showing the surface texture created during the printing process; (**d**) a thin-wall sample printed with a high aspect ratio.

**Figure 5 materials-17-02188-f005:**
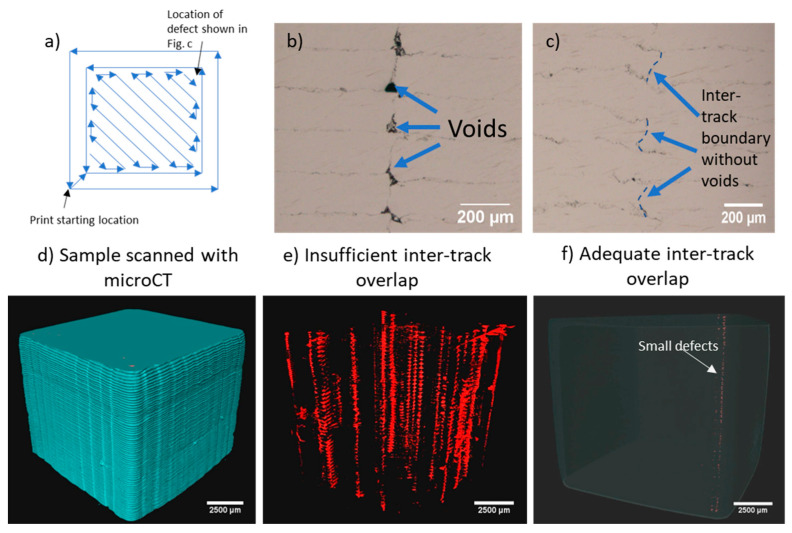
(**a**) The tool pathing strategy used for printing the components. The strategy uses 2 outer walls and a +45°/−45° infill. The outer walls were printed first, followed by the infill. (**b**) Optical image of the cross section of a printed component with insufficient overlap (inter-track distance of 0.7 mm); (**c**) optical image of the cross section of a printed component with sufficient overlap (inter-track distance of 0.6 mm); (**d**) geometry of an as-printed sample scanned with micro-CT. (**e**) Internal defect structure in a sample printed with insufficient overlap between adjacent tracks (inter-track distance of 0.7 mm). The red region in this figure shows the size and shape of the defects. (**f**) Internal defect structure in a sample printed with adequate overlap (inter-track distance of 0.7 mm). The red defect region is significantly reduced with optimized process parameters.

**Figure 6 materials-17-02188-f006:**
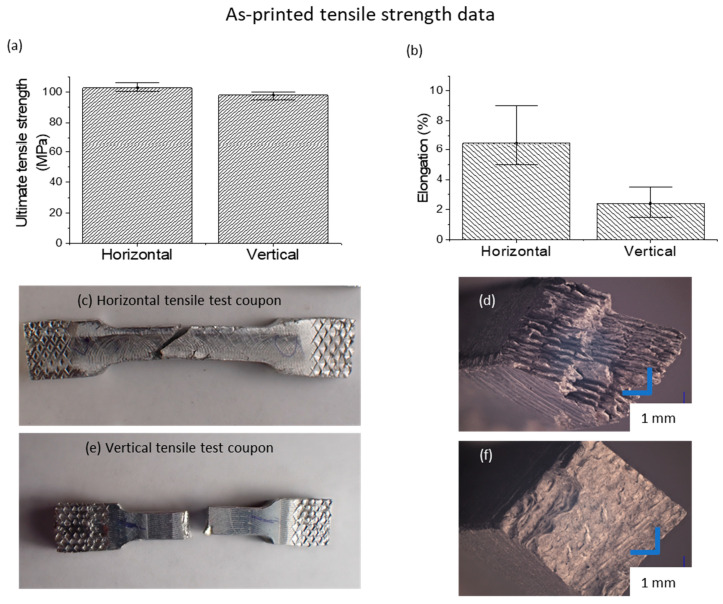
(**a**,**b**) Ultimate tensile strength (UTS) and elongation values for the horizontal and vertical test coupons, respectively; (**c**) optical image of a horizontal test coupon after tensile testing; (**d**) optical image of the fracture surface of a horizontal test coupon after tensile testing; (**e**) optical image of a vertical test coupon after tensile testing; (**f**) optical image of the fracture surface of a vertical test coupon after tensile testing.

**Table 1 materials-17-02188-t001:** Table summarizing the process parameters used during the printing of the Al6061 articles shown in Figure 2, Figure 3, Figure 4, Figure 5 and Figure 6.

Process Parameter	Value
Tool oscillation frequency	40 KHz
Tool oscillation displacement	1 μm peak-to-peak
Voxel and layer height	0.12 mm
Step size	1 mm
Inter-track distance	0.6 mm and 0.7 mm

**Table 2 materials-17-02188-t002:** Relevant process parameters for the energy calculations.

Bulk modulus (*G*), GPa	70
Voxel X–Y plane area (*A*), m^2^	4.8 × 10^−7^
Oscillatory shear displacement (*a*), m	1 × 10^−6^
Oscillatory shear displacement frequency (*f*), Hz	40 × 10^3^
Voxel compression time (*t*), s	0.023
Tool load during compression ($F_{static}$), N, experimentally measured for Al6061-O	100
Feedstock wire diameter (*d*), m	3.5 × 10^−4^
Voxel height (*h*), m	1.2 × 10^−4^

## Data Availability

Data will be provided on request.

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
