# Peer review of "The Production of Three-Dimensional Metal Objects Using Oscillatory-Strain-Assisted Fine Wire Shaping and Joining"

_materials, 2024, doi:10.3390/ma17102188_

Round 1

Reviewer 1 Report

Comments and Suggestions for Authors

This manuscript showed a 3D print method for metal production by using oscillatory strain assisted fine wire shaping and joining. Overall, the manuscript is well written. I therfore recommend to accept this manuscript after a minor revision. The specific comments are given below. 

1. The unique contribution of this study should be highlighted in section Introduction. Especially the progressiveness of this work compared with traditional 3D metal print.

2. In your supplementary, I noticed the triangular surface is very smooth; while the other seeable surface is rough. What is the reason? Is it possible to make every surface smooth?

3. The citation of the figures and tables should be update. It showed "Error! Reference source not found" throught.

4. The layout of Fig. 6 (a) and (b) is poor, please revise it. 

5. After all the experiments, a recommended 3D print flow should be summarized.

Author Response

This manuscript showed a 3D print method for metal production by using oscillatory strain assisted fine wire shaping and joining. Overall, the manuscript is well written. I therfore recommend to accept this manuscript after a minor revision. The specific comments are given below. 

Authors’ response: The authors would like to thank the reviewer for their valuable comments. Specific comments by the reviewer have been addressed below.

  1. The unique contribution of this study should be highlighted in section Introduction. Especially the progressiveness of this work compared with traditional 3D metal print.

Authors’ response: The 3d printing method demonstrated in this manuscript has 3 main advantages over existing metal 3d printing processes: 1) It is more energy efficient than existing melt-fusion based processes, 2) it enables metal 3d printing of hard-to-weld alloys like Al6061, 3) it can print net shape components without the need of a post-processing operation. These advantages have been highlighted in lines 106-113 of the Introduction section. These lines from the manuscript have been added below for the reviewer’s convenience.

“In this manuscript, we demonstrate a novel method, the Resonance Assisted Deposition (RAD), for metal 3D printing by simultaneously shaping and joining a fine wire continuously into metal voxels in solid state. It is shown that this method is significantly more energy efficient than using heating, melting, and solidification. The method has the ability to print net shape components from hard-to-weld alloys Al6061 and the ability to print components with high aspect ratio. Because of these specific advantages, this novel method has the potential to the method-of-choice in applications where existing metal 3D printing are lacking.”

  1. In your supplementary, I noticed the triangular surface is very smooth; while the other seeable surface is rough. What is the reason? Is it possible to make every surface smooth?

Authors’ response: The surfaces of all the side walls of a component printed with RAD method have “scalloping” on them very similar to that seen in Fused Filament Fabrication polymer printing method. The apparent smoothness of one of the faces in the Supplemental video is actually an optical illusion due to the angle of the camera to the apparently smooth surface. All side surfaces have the same surface roughness in components printed with RAD. Further explanation of the surface morphology of printed components is given in section 3.2 of the manuscript. The has been added below for the reviewer’s convenience:

“Top surface of printed metal parts (surface orthogonal to the print direction) has different surface morphology as compared with the side surfaces parallel to the build direction. The top surface morphology is dictated by the tooling surface in contact with the metal voxel during each compression cycle, as well as the material flow and track-to-track overlaps in the tool path. The bottom surface morphology of the build is determined by the joint strength between the first layer of the printed part and the build plate, and how it is separated from the build plate. The side surfaces have a characteristic “scalloping” effect similar to what is typically seen in the Fused Filament Fabrication polymer 3D printing method as large amounts of material strain is experienced by the feedstock material during the printing process.  In the RAD process, the ratio of voxel height (also layer height) to feedstock diameter is between 0.4.  While the exact ratio is dictated by the amount of compression required in each compression cycle to establish full space filling between adjacent tracks and proper joining of interfaces, this ratio also determines the extent of the scallops on the side surfaces.  The larger the layer height, the higher the waviness on the sidewalls. Within the feasible parameter range, the resulting roughness values of top surfaces are typically around 10-20 micrometers Ra, while the side surfaces are in the range of 15-25 micrometers Ra.”

  1. The citation of the figures and tables should be update. It showed "Error! Reference source not found" throught.

Authors’ response: Authors had taken great care in making sure that the formatting of the manuscript uploaded to the system was accurate. It seems that when the system automatically converted the word document to pdf format, citation of the figures and tables incurred errors. The authors would like to apologize for the inconvenience. The latest version of the manuscript has been thoroughly checked for any such formatting errors. 

  1. The layout of Fig. 6 (a) and (b) is poor, please revise it. 

Authors’ response: The layout of Fig 6a and 6b has been revised. The figures have been spaced appropriately to ensure that it is easier for the reader to read them. The new figure has been added below (in the word document for the response) for the reviewer’s convenience:

  1. After all the experiments, a recommended 3D print flow should be summarized.

Authors’ response: the authors interpret the reviewer’s comment as describing the workflow. A section describing such as been added.

“To implement the RAD (Resonance Assisted Deposition) method into a system to produce a 3D metal object, a 3-axis motion system was used. A schematic of the experimental setup is shown in Figure 1a. The process begins with slicing of a CAD model into layers and then subsequent generation of toolpaths to fill the layers. The toolpaths are then sent to the motion system for execution.”

Reviewer 2 Report

Comments and Suggestions for Authors

The research showed the applicability of the resonance assisted deposition technology as technology used for 3D Printing.

The decision to reject the paper was made due to following points:

1) references to figures were erroneous

2) figures were not sorted according to their sequence in the text. Furthermore figures should be placed where mentioned in the text. In this manuscript all figures were put in the same place

3) no logic behind which figures are in the supplementary material and which are in th original text

Based on this simple formatting issues the authors were not able to manage, the reviewer came to the decision that this manuscript is not suitable for publication! However the reviewer would like to make some additional comments on the research work

4) not all techniques used in the manuscript were described properly such as thermocamera and thermocouples

5) what is the spacial resolution of the thermocamera? Is this sufficient enough to measure the temperature of this process? Timeline resolution for the thermocouple?

6) SEM images of the microstructure and fracture surface is missing. This should be easily available when TEM is also used and should be the first step of investigation

6) proper description of the cross section is missing => obviously there are interlayer lack of fusions visible. Should be clarified by SEM

7) theory of lattice defect diffusion should be discussed in more depth. Transferability of Ni-Al TEM investigations need also better discussion as Ni has very different plasticity behaviour compared to Al

8) 99.95% density is very unrealistic for the cross sections shown. Line porosity has to be discussed as this is worst case porosity for mechanical properties

9) chapter 3.3 is missing proper source declaration

10) line 280: definition of variables is missing

11) comparison to Al6061 -T6 should be drawn. Tensile testing of heat treated specimen would be more valuable compared to O temper

Comments on the Quality of English Language

O.k.

Author Response

The research showed the applicability of the resonance assisted deposition technology as technology used for 3D Printing.

The decision to reject the paper was made due to following points:

1) references to figures were erroneous

2) figures were not sorted according to their sequence in the text. Furthermore figures should be placed where mentioned in the text. In this manuscript all figures were put in the same place

3) no logic behind which figures are in the supplementary material and which are in th original text

Based on this simple formatting issues the authors were not able to manage, the reviewer came to the decision that this manuscript is not suitable for publication! However the reviewer would like to make some additional comments on the research work.

Authors’ response: Authors had taken great care in making sure that the formatting of the manuscript uploaded to the system was accurate. It seems that when the system automatically converted the word document uploaded by the authors to the journal specific word and pdf format, citation of the figures and tables incurred errors. Along with this, the position of the figures in the manuscript also changed. The authors would like to apologize for the inconvenience but would also like to assert that rejecting years’ worth of work just based on system generated formatting errors is extremely harsh. The latest version of the manuscript has been thoroughly checked for any such formatting errors.  The authors would like to request the reviewer to reconsider their rejection.

4) not all techniques used in the manuscript were described properly such as thermocamera and thermocouples

Authors’ response: Thermal camera and thermocouple measurements were performed in a previous work published by the authors. The manuscript has been updated to clarify and explicitly mention this fact and appropriate citation to the previous work by the authors has been added. The relevant lines from the manuscript have been added below for the reviewers’ convenience:

“In the work previously published by the authors [14] regarding a precursor method to the RAD technique that uses the same deposition method, thermocouple and high-speed infra-red camera measurements (Supplemental Figure 1) showed that the temperature rise actually measured during the deposition process is of the order of 5-10 ⁰C.”

5) what is the spacial resolution of the thermocamera? Is this sufficient enough to measure the temperature of this process? Timeline resolution for the thermocouple?

Authors’ response: Thermal camera and thermocouple measurements were performed in a previous work published by the authors. The manuscript has been updated to clarify and explicitly mention this fact and appropriate citation to the previous work by the authors has been added. A FLIR A6751 camera was used for IR imaging of the interface. The camera has a pixel size of 15 µm square. The imaging frequency was 125.6 Hz, which means that the temperature was recorded every ~8 milliseconds during the process. The data acquisition frequency for the thermocouple measurements was 250 Hz.

6) SEM images of the microstructure and fracture surface is missing. This should be easily available when TEM is also used and should be the first step of investigation.

Authors’ response: The authors believe the current content in the manuscript is optimal which has an intent of introducing this new metal additive manufacturing technique. Optical images of the fracture surface are given in Figure 6d and f. These images are sufficient to provide evidence for the analysis and claims about fracture behavior made in the manuscript. A more detailed analysis on microstructure and fracture behaviors will be thoroughly reported in follow up manuscripts which will then be reported with SEM images of microstructure and fracture surfaces.

7) proper description of the cross section is missing => obviously there are interlayer lack of fusions visible. Should be clarified by SEM

Authors’ response: The interlayer “lack of fusions” that are visible in Figure 5b and c are formed due to entrapped native oxides. These native oxides are shown in the TEM image in Figure 3a. This has been now clarified in the manuscript. It has also been explained in the Mechanical properties section (section 3.4) that the lower elongation can be attributed to the trapped native oxides and easier crack propagation. The relevant snippets from the manuscript regarding native oxides and their effect on mechanical properties are given below for the reviewer’s convenience:

“…Native oxides that break due to the initial friction get trapped at the interface during the deposition process and can also be seen in Figure 3a and c.”

(New addition) ”…In the cross-sections shown in Figure 5b and c, darker boundaries clearly delineate the interlayer interface. This is because of the native oxides discussed earlier (Figure 3a) that get trapped at the interlayer interface preventing nascent metal contact.”

“…The lower elongation can be attributed to the trapped native oxides discussed in the previous section which result in easier crack propagation in the regions of the interface where oxides have been trapped.”

8) theory of lattice defect diffusion should be discussed in more depth. Transferability of Ni-Al TEM investigations need also better discussion as Ni has very different plasticity behaviour compared to Al

Authors’ response: Lattice defect diffusion (or vacancy diffusion) in metals happens when atoms jump from their current lattice position to take the place of a defect. Lattice defect diffusion itself is a very basic phenomenon and should not require any further explanation. What is unique to the RAD technique is the enhancement to this phenomenon which enables increased diffusion across the interfaces. It is hypothesized that this enhancement is provided by the increased defect density gradient across the voxel-voxel interface. The higher defect density region that increases the defect density gradient is shown in the TEM images (Figure 3).

The reviewer is correct to point out that the quantitative analysis of the Al-Ni system will not directly apply to the Al-Al system because the diffusivity of Al in Ni is different from the diffusivity of Al in Al. The main objective of the Al-Ni investigation was to provide evidence of enhanced diffusion across the interface during the deposition process in RAD. To conduct this investigation, it was essential to use some material other than Al as a substrate so that the concentration of the two metals across the interface could be measured using EDS. The authors would like to clarify that the plasticity behavior of Ni, as mentioned in the question by the reviewer, should have nothing to do with the diffusion since Nickel is being used as a substrate in the investigation and does not deform significantly (if at all) during the deposition process.

Explanation to clarify the rationale behind the use of Al-Ni system has been added to the manuscript. The relevant addition is given below for the reviewer’s convenience:

“The quantitative analysis of the Al-Ni system will not directly apply to the Al-Al system typically seen in RAD because the diffusivity of Al in Ni is different from the diffusivity of Al in Al. The main objective of the Al-Ni investigation was to provide evidence of enhanced diffusion across the interface during the deposition process in RAD. To conduct this investigation, it was essential to use some material other than Al as a substrate so that the concentration of the two metals across the interface could be measured using EDS.

9) 99.95% density is very unrealistic for the cross sections shown. Line porosity has to be discussed as this is worst case porosity for mechanical properties.

Authors’ response: As indicated in section 2.2, the scans for micro-CT analysis were performed at a resolution of 6.95 μm. Any defect smaller than that would not appear in the micro-CT scan. Also, as discussed in response to earlier question, the darker lines at the inter-layer interfaces are due to trapped native oxides which are shown in the TEM image in Figure 3a. The effect of these native oxides on mechanical properties has been discussed in the “Mechanical Properties” section (section 3.4). The relevant snippets from the manuscript regarding native oxides and their effect on mechanical properties are given below for the reviewer’s convenience:

“…Native oxides that break due to the initial friction get trapped at the interface during the deposition process and can also be seen in Figure 3a and c.”

(New addition) ”…In the cross-sections shown in Figure 5b and c, darker boundaries clearly delineate the interlayer interface. This is because of the native oxides discussed earlier (Figure 3a) that get trapped at the interlayer interface preventing nascent metal contact.”

“…The lower elongation can be attributed to the trapped native oxides discussed in the previous section which result in easier crack propagation in the regions of the interface where oxides have been trapped.”

10) chapter 3.3 is missing proper source declaration

Authors’ response: The authors appreciate this comment. However, calculations of mechanical work done in the manuscript follows the foundational relationship of work = ½ * stress * strain. The “1/2 “ was not used as the calculation provides an upper bound estimate.  This estimative calculation is the author’s work based on foundational Newtonian Mechanics.

11) line 280: definition of variables is missing

Authors’ response: The variable definitions have been added to the manuscript. The equation and the associated equation are given below for the reviewer’s convenience:

where : force required to shear the voxel in the direction of oscillatory strain,

: the oscillatory shear displacement,

: force required to compress the voxel in z-direction,

: compression of the voxel to its layer height.

11) comparison to Al6061 -T6 should be drawn. Tensile testing of heat treated specimen would be more valuable compared to O temper

Authors’ response: The authors appreciate the reviewer’s comment. However, the authors believe the current content in the manuscript is optimal which has an intent of introducing this new metal additive manufacturing technique.  Effect of post-fabrication heat treatments and processes will be thoroughly reported in follow up manuscripts.

Reviewer 3 Report

Comments and Suggestions for Authors

This paper seems to be a study of the additive manufactured 6061-O aluminium alloy material with a new method. The characterisation of the materials microstructure and mechanical properties seems to be correctly carried out. However, this paper shows several deficiencies on the experimental setup, design of experiments, discussion, reference and conclusions. Thus, this paper might be considered to be published after improving it with the next comments. 

"We" word should be avoided because this reduces the objectivity of the paper.

The mechanical properties and microstructure should be compared with other samples as wrought or/and laser additive manufactured samples to understand why the samples are better. 

The assertion " Laser 60 powder bed fusion for example, uses solid-state fiber lasers which operate at around 20% 61 energy efficiency." is fake, the efficiency of the laser depend of the material type.

The error "Error! Reference source not found" should be corrected because this hinders the understanding of the paper.

the paragraph of the page 3 from 124 to 133 should be moved to results and discussion because these are results. 

The supplier and the type of the material should be added in 2.Methodology part because these are essential part of the experimental setup. 

Figure 2-6 should be in results and discussion because these are results. 

The model, supplier and testing setup of the tensile testing device should be added in 2.Methodology part because these are essential part of the testing conditions. 

The data from EDS is not provided in the results and discussion. This should be included because these are part of your results.

The details of the TEM and EDS (type of electron, acceleration voltage, tension current and spot size) should be provided in 2.Methodology part because the results can be modified for these. 

TEM images of the specimens should be included and commented in the Results and discussion part because these are the results. 

The type, model, supplier and analysis conditions of the device employed to measure the average roughness should be included in 2.Methodology part owing to this an essential part.

If the equations were not developed by  the researchers, these should be referenced to avoid plagiarism complaint.

All symbol of the equation should be defined to avoid confusions. 

The data of this assertion "On the other hand, the overall fracture behavior of built material exhibits less ductility when the tensile load is applied along the build direction. " (page 11 334-336) are absent, please include them. 

In the conclusions, the tensile testing results are ignored, this should be included. 

My specific comments are the following:

To add reference in line 41 of page 1 "facturing processes [REF]. "

To insert reference in line 66 of page 2 "forms the melt pool to accomplish shaping the joining [REF]. Additionally, the inefficiencies in"

To include reference in line 92 of page 2 "defect regions is more effective [REF]. As demonstrated in Legro’s work where bulk diffusion of"

To add reference in line 105 of page 3 "space and time, an entirely new 3D printing approach can emerge [REF]."

To define the acronym RAD in line 105 of page 3 "To implement the RAD method into a system to produce a 3D metal object, a 3-axis"

To include reference in line 251 of page 9 "to be ranging from ~320 ⁰C to 420 ⁰C [REF]. These temperatures values are much higher than the"

To incorporate reference in line 299 of page 10 "used [REF]."

To add reference in line 331 of page 11 "easier crack propagation in the regions of the interface where oxides have been trapped [REF]."

Author Response

  • This paper seems to be a study of the additive manufactured 6061-O aluminium alloy material with a new method. The characterisation of the materials microstructure and mechanical properties seems to be correctly carried out. However, this paper shows several deficiencies on the experimental setup, design of experiments, discussion, reference and conclusions. Thus, this paper might be considered to be published after improving it with the next comments. 

Authors’ response: The authors would like to thank the reviewer for their valuable comments. Specific comments by the reviewer have been addressed below.

  • "We" word should be avoided because this reduces the objectivity of the paper.

Authors’ response: The manuscript has been revised to avoid the word “We”.

  • The mechanical properties and microstructure should be compared with other samples as wrought or/and laser additive manufactured samples to understand why the samples are better. 

Authors’ response: The mechanical properties have been compared to the feedstock material in the manuscript. It was found that the ultimate tensile strength was close to 75% of that of the raw Al-6061 O-annealed feedstock. This information has been added to the manuscript. Please refer below:

“The ultimate tensile strength of as-printed samples was found to be close to 75% of that of the raw Al-6061 O-annealed feedstock.”

The authors would like to clarify that they do not claim anywhere in the manuscript that the samples are better than wrought in terms of strength. Also, comparison with laser additive manufactured samples is not possible since Al-6061 without any special feedstock can not be fabricated by laser additive manufacturing (or any other melt-fusion based additive manufacturing). Currently, using a special feedstock that alters the composition of the alloy is being employed by some researchers and manufacturers to produce Al-6061 components with laser additive manufacture. But comparison with samples manufactured with Al-6061 with altered composition would not be meaningful.

  • The assertion " Laser 60 powder bed fusion for example, uses solid-state fiber lasers which operate at around 20% 61 energy efficiency." is fake, the efficiency of the laser depend of the material type.

Authors’ response: The authors appreciate the reviewer’s comment. We have revised the statement. 

“Laser powder bed fusion for example, uses solid-state fiber lasers which operate at low energy efficiency.”

  • The error "Error! Reference source not found" should be corrected because this hinders the understanding of the paper.

Authors’ response: Authors had taken great care in making sure that the formatting of the manuscript uploaded to the system was accurate. It seems that when the system automatically converted the word document to pdf format, citation of the figures and tables incurred errors. The authors would like to apologize for the inconvenience. The latest version of the manuscript has been thoroughly checked for any such formatting errors. 

  • the paragraph of the page 3 from 124 to 133 should be moved to results and discussion because these are results. 

Authors’ response: Paragraph of page 3 from 124 to 133 describes the steps that take place during the RAD printing process. Hence, the authors believe that it belongs in the Methodology section. The product of these steps is a printed component. The images and associated analysis of these components is in the Results and Discussion section.

  • The supplier and the type of the material should be added in 2.Methodology part because these are essential part of the experimental setup. 

Authors’ response: The following line has been added to the manuscript based on the reviewer’s request.

“All articles displayed in Figure 2 to 6 were printed with annealed Al6061-O feedstock wire with 0.350 mm diameter supplied by California Fine Wire Company. The feedstock wire had a tensile strength of 140 MPa and an elongation of 17.1%.”

  • Figure 2-6 should be in results and discussion because these are results. 

Authors’ response: This is another piece of formatting that incurred errors when the system converted the uploaded manuscript to journal specific formatting. This location of the figures in the manuscript has been now corrected.

  • The model, supplier and testing setup of the tensile testing device should be added in 2.Methodology part because these are essential part of the testing conditions. 

Authors’ response:

  • The data from EDS is not provided in the results and discussion. This should be included because these are part of your results.

Authors response: The EDS data has been provided in the Supplemental Material. This is because the EDS data provides supplemental evidence to the hypothesis that oscillatory shear strain enhances the interlayer diffusion. These EDS results feed into the additional details about the calculations performed to calculate the effective temperature values necessary if equivalent thermal diffusion were to take place. The fact that the temperature rise during the process is much smaller than the effective temperature required for thermal diffusion supports the hypothesis that oscillatory shear strain indeed enhances diffusion across the interface.

  • The details of the TEM and EDS (type of electron, acceleration voltage, tension current and spot size) should be provided in 2.Methodology part because the results can be modified for these.

 Authors’ response:  For TEM imaging, an accelerating voltage of 200 kV and a spot size of 1 was used. This information has been added to the manuscript. The authors would like to clarify that these parameters are very machine specific and will not change the results but might affect the imaging quality.

EDS was performed in STEM mode at an accelerating voltage of 200 kV, a spot size of 2 (which also controls the current) and a step size of 5 nm. This information has been added to the manuscript. Authors would like to clarify that the EDS results provided in the manuscript show relative composition of the elements (Al and Ni in our case) which will not change with a change in these parameters, especially when the composition of the two elements is in such a high concentration and the accelerating voltage of 200 kV is much higher than the K-alpha energy for Al and Ni which can be a constraint in SEMs with lower accelerating voltages.

  • TEM images of the specimens should be included and commented in the Results and discussion part because these are the results. 

Authors’ response: As mentioned earlier, this error was due to automatic conversion of the uploaded manuscript to journal specific formatting. TEM images of the specimen have been moved to the Results and Discussion Section in the revised version of the manuscript.

  • The type, model, supplier and analysis conditions of the device employed to measure the average roughness should be included in 2.Methodology part owing to this an essential part.

Authors’ response: A Mitutoyo Surftest portable surface roughness tester was used to measure the average roughness. The following line has been added to the manuscript:

“To analyze the average surface roughness of printed components, a Mitutoyo Surftest portable surface roughness tester was used. The surface roughness was measured on surfaces parallel to the X-Y and X-Z (or Y-Z) planes.”

  • If the equations were not developed by  the researchers, these should be referenced to avoid plagiarism complaint.

Authors’ response: The equations are very specific to the RAD technique and were developed by the authors.

  • All symbol of the equation should be defined to avoid confusions. 

Authors’ response: The variable/symbol definitions have been added to the manuscript.

  • The data of this assertion "On the other hand, the overall fracture behavior of built material exhibits less ductility when the tensile load is applied along the build direction. " (page 11 334-336) are absent, please include them. 

Authors’ response: In Figure 6b, elongation of the vertical samples (samples for which load is applied along the build direction) is much lower than that of the horizontal samples (samples for which load is applied perpendicular to the build direction). Also, the fracture surface shown in Figure 6b does not show any plastic deformation. Both these details show that the overall fracture behavior of built material exhibits less ductility when the tensile load is applied along the build direction. To clarify this, the statement has been modified in the manuscript as given below:

“On the other hand, the overall fracture behavior of built material exhibits less ductility when the tensile load is applied along the build direction exhibiting a brittle fracture as is evident in the fracture surface shown in Figure 6f which shows no evidence of plastic deformation but just a clear separation at the interlayer interface.”

  • In the conclusions, the tensile testing results are ignored, this should be included. 

Authors’ response: The following line about tensile testing results has been added to the Conclusion section.

“Strength of as-built parts are within 75% of that of the feedstock material with the elongation being lower for parts when tensile load is applied in the build direction.”

My specific comments are the following:

  • To add reference in line 41 of page 1 "facturing processes [REF]. "

Authors’ response: Reference has been added.

  • To insert reference in line 66 of page 2 "forms the melt pool to accomplish shaping the joining [REF]. Additionally, the inefficiencies in"

Authors’ response: Reference has been added.

  • To include reference in line 92 of page 2 "defect regions is more effective [REF]. As demonstrated in Legro’s work where bulk diffusion of"

Authors’ response: Reference has been added.

  • To add reference in line 105 of page 3 "space and time, an entirely new 3D printing approach can emerge [REF]."

Authors’ response: This statement is about the work done by the authors in the manuscript. It tries to indicate that with metal shaping and joining, an entirely new 3d printing approach can emerge as demonstrated in the manuscript. The statement is followed by a paragraph on how the authors have demonstrated the emergence of such a 3d printing approach. It is precursor to the claims in the manuscript. For this reason, a reference has not been added.

  • To define the acronym RAD in line 105 of page 3 "To implement the RAD method into a system to produce a 3D metal object, a 3-axis"

Authors’ response: The acronym has been defined in the line in the revised manuscript.

  • To include reference in line 251 of page 9 "to be ranging from ~320 ⁰C to 420 ⁰C [REF]. These temperatures values are much higher than the"

Authors’ response: A reference to the Supplemental Material has been added here where the calculations of the effective temperature required for equivalent diffusion have been performed.

  • To incorporate reference in line 299 of page 10 "used [REF]."

Authors’ response: This is an assumption made by the authors in their calculations. So no reference has been added.

  • To add reference in line 331 of page 11 "easier crack propagation in the regions of the interface where oxides have been trapped [REF]."

Authors’ response: This is an observation made by the authors in their calculations. So no reference has been added.

Reviewer 4 Report

Comments and Suggestions for Authors

Dear Authors,

The paper entitled” Production of Metal 3D Objects Using Oscillatory Strain 2 Assisted Fine Wire Shaping and Joining” is focused on the 3D metal printing which represents an actuality research subject.

The authors described the Resonance Assisted Deposition (RAD) technology, as a new method for producing 3D metal parts.

There are some minor observations, namely:

- Please specify in the abstract the RAD technology

- Line 115 „Error! Refer- 115 ence source not found” and almost all the references in the manuscript

- Please specify the diameter of the nozzle.

- Try to add more references 

Best regards!

Author Response

The paper entitled” Production of Metal 3D Objects Using Oscillatory Strain 2 Assisted Fine Wire Shaping and Joining” is focused on the 3D metal printing which represents an actuality research subject.

The authors described the Resonance Assisted Deposition (RAD) technology, as a new method for producing 3D metal parts.

Authors’ response: The authors would like to thank the reviewer for their valuable comments. Specific comments by the reviewer have been addressed below.

There are some minor observations, namely:

- Please specify in the abstract the RAD technology

Authors’ response: The abstract has been modified to specify RAD technology in the abstract. The relevant addition has been given below for the reviewer’s convenience:

            “The manuscript demonstrates that by using the physical phenomenon of lowered yield stress of metals and enhanced diffusion in the presence of low amplitude high frequency oscillatory strain, metal shaping and joining can be performed energy efficiently. The two performed simultaneously enable a metal additive manufacturing process, Resonance Assisted Deposition (RAD), that has several unique capabilities like the ability to print net shape components from hard-to-weld alloys Al6061 and the ability to print components with very high aspect ratio. We show the process capabilities by printing solid components in aluminum-based metal alloys.”

- Line 115 „Error! Refer- 115 ence source not found” and almost all the references in the manuscript

Authors’ response: Authors had taken great care in making sure that the formatting of the manuscript uploaded to the system was accurate. It seems that when the system automatically converted the word document to pdf format, citation of the figures and tables incurred errors. The authors would like to apologize for the inconvenience. The latest version of the manuscript has been thoroughly checked for any such formatting errors. 

- Please specify the diameter of the nozzle.

Authors’ response: The nozzle (or shear-strain transfer tool) has a hole of 0.5 mm diameter. This information has been added to the manuscript. The addition has been given below for the reviewer’s convenience:

“Solid wire feedstock is made to pass through a hollow cylindrical shear-strain transfer tool that has a hole of 0.5 mm diameter.”

- Try to add more references 

Authors’ response: More references have been added as per the reviewer’s request.

Best regards!

Round 2

Reviewer 3 Report

Comments and Suggestions for Authors

This paper is a study of the additive manufactured 6061-O aluminium alloy material with a new method. The characterisation of the materials microstructure and mechanical properties was correctly carried out.  Thus, this paper can be published after considering it with the next comments. 

The supplier and model of the powermeter should be added in 2.Methodology part because the readers can want to know a good supplier and model. 

My specific comment are the follwing:

To add reference in "terface [REF]. This is because of the native oxides discussed earlier (Figure 3a) that get trapped"

To include reference in "propagation in the regions of the interface where oxides have been trapped [REF]. In the frac-"

Author Response

  • This paper is a study of the additive manufactured 6061-O aluminium alloy material with a new method. The characterisation of the materials microstructure and mechanical properties was correctly carried out.  Thus, this paper can be published after considering it with the next comments. 

Authors’ response: The authors would like to thank the reviewer for their valuable comments. Specific comments by the reviewer have been addressed below.

  • The supplier and model of the powermeter should be added in 2.Methodology part because the readers can want to know a good supplier and model. 

Authors’ response: The following details have been added to the manuscript in response to the reviewer’s comments:

“The oscillatory strain generator, motion and control systems (Figure 1) used to demonstrate metal 3D printing using RAD was outfitted with a power meter (KW47-US) that measures the power draw of the entire system during printing.”

  • To add reference in "terface [REF]. This is because of the native oxides discussed earlier (Figure 3a) that get trapped"

Authors’ response: The specific sentence refers to the observations made by the authors and hence no reference has been added.

  • To include reference in "propagation in the regions of the interface where oxides have been trapped [REF]. In the frac-"

Authors’ response: This is a conclusion drawn by the authors. Horizontal and vertically printed samples show similar UTS but the elongation of vertical samples is lower. The fracture surface for vertically printed samples shown in Figure 6f shows an interface devoid of much plastic deformation. Both of these facts in conjunction suggest that the crack propagation is much faster in vertically printed samples and it is the authors’ claim that this is because of the trapped oxides that inhibit nascent metal contact and diffusion across the interface.

Since this is also a conclusion drawn by the authors, no reference has been added.
